# Perspectives of healthcare professionals in England on falls interventions for people with dementia: a qualitative interview study

Clare Burgon,[1,2] Janet Darby,[1] Kristian Pollock,[2] Veronika van der Wardt,[1] Tamsin Peach,[3] Lyndsay Beck,[4] Pip Logan,[1] Rowan H Harwood[1,2,5]

¹Division of Rehabilitation and Ageing, School of Medicine, University of Nottingham, Nottingham, UK
²School of Health Sciences, University of Nottingham, Nottingham, UK
³Nottingham CityCare Partnership, Nottingham, UK
⁴Nottinghamshire Healthcare NHS Trust, Nottingham, UK
⁵Nottingham University Hospitals NHS Trust, Health Care of Older People, Nottingham, UK

**Correspondence to**
Clare Burgon;
clare.burgon@nottingham.ac.uk

## ABSTRACT

**Objective** To explore the experiences of healthcare professionals working in falls prevention and memory assessment services in providing assessments and interventions for falls risk reduction in people with dementia.

**Design** This is a qualitative study using 19 semistructured interviews. Interviews were analysed through thematic analysis.

**Setting** Community-based falls and memory assessment services in the East Midlands, UK.

**Participants** Nurses (n=10), physiotherapists (n=5), occupational therapists (n=3) and a psychiatrist (n=1).

**Results** Three substantive themes were identified: challenges posed by dementia, adaptations to make falls prevention appropriate for people with dementia and organisational barriers. Patients' poor recall, planning and increased behavioural risk associated with dementia were key problems. Healthcare professionals provided many suggestions on how to overcome these challenges, such as adapting exercise interventions by using more visual aids. Problems associated with cognitive impairment created a need for additional support, for instance longer interventions, and supervision by support workers, to enable effective intervention, yet limited resources meant this was not always achievable. Communication between mental and physical health teams could be ineffective, as services were organised as separate entities, creating a reliance on third parties to be intermediaries. Structural and organisational factors made it difficult to deliver optimal falls prevention for people living with dementia.

**Conclusions** Healthcare professionals experience challenges in providing falls prevention to people with dementia at the individual and organisational levels. Interventions can be adapted for people with dementia, but this requires additional resources and improved integration of services. Future research is needed to develop and test the effectiveness and cost-effectiveness of such services.

## BACKGROUND

Around one in three adults over 65 fall each year,[1 2] with 12% of older people falling multiple times.[3] Falls are associated with injury, loss of confidence, restriction of activity, nursing home admission[4] and worse quality of life 6 years later.[5] Falls requiring hospital admission in those over 75 years old result in 6% mortality and 14% hospital readmission within 30 days.[6] Falls present a significant economic burden, costing the National Health Service more than £2.3 billion per year.[7]

People with dementia are twice as likely to experience a fall, even at the very earliest stages,[8–10] are more likely to be hospitalised due to fall-related fractures,[11] and have higher risks of complications following a fracture[12] than people without cognitive impairment.

The UK National Institute for Health and Care Excellence recommends multifactorial interventions to prevent falls, which involve the assessment and correction of home hazards and visual impairment, reviewing medications, and prescription of exercises to improve strength and balance.[7] In the UK falls prevention is often delivered by community-based, multidisciplinary falls prevention teams. A Cochrane review demonstrated that multifactorial interventions decrease falls rates, with strength and balance training being particularly effective.[13] The effectiveness of standard falls prevention for people with dementia is inconclusive, however.[14] It has been argued that dementia

is associated with distinct risk factors for falls, not targeted by standard interventions, thereby limiting potential effectiveness,[15] and that attention to uptake of, and adherence to, falls prevention programmes has been inadequate.[16 17] Trials of exercise-based interventions involving people with dementia suggest that falls can be reduced through strength and balance training, but that interventions require additional support or adaptation.[18] Practice guidelines are equivocal on the best approach for people with dementia at risk of falling.[19]

**Strengths and limitations of this study**

► We sought the perspectives and experiences of healthcare professionals from different real-world services.
► We recruited participants from one geographical area, so findings may not be transferable to other localities.
► The interviewer was an occupational therapist with a background in falls prevention, which may have influenced participants, encouraging openness, but potentially leading to failure to recognise tacit assumptions.

There is a pressing need to develop and test falls prevention interventions that are designed specifically for people with dementia living in the community.[20] This report is from a study which aimed to do this.[21–23] A recommended first step in intervention development is to consult stakeholders to help understand the problem, challenges to intervention and the context in which it would be delivered.[24] Stakeholders in this case include people living with dementia, their families and other carers, and healthcare professionals (HCPs), such as physiotherapists, occupational therapists and nurses, involved in their care. We previously reported the views of people living with dementia and their family carers.[22] HCPs can offer additional insights into problems and potential solutions, the delivery and acceptability of interventions, and the practical constraints on clinical services.[25] HCPs also have a role in influencing patients' opinions on falls and uptake of interventions.[26]

This study aimed to understand the services in which the HCPs worked, explore their perspectives and experiences of working with, and delivering interventions to, people with dementia living in the community, and gather recommendations for the design and delivery of effective and acceptable falls prevention techniques in this patient group.

## METHODS
### Design
Individual, face-to-face, semistructured interviews were undertaken, as part of a qualitative descriptive approach,[27] in order to provide evidence to inform the development of a falls prevention intervention for people with dementia.[21 22] Inclusion criteria were (1) being an HCP and (2) working within a memory assessment service (a dementia diagnostic service run by mental health providers and overseen by consultant psychiatrists) or delivering falls prevention services, including to people with diagnosed or undiagnosed dementia.

### Patient involvement
A patient and public involvement (PPI) advisory panel helped develop and oversee the initial research questions and study application. The protocol was adapted based on feedback from lay reviewers, who gave strong support for the interview studies. A PPI coapplicant regularly contributed to the study at management meetings.

### Participants and setting
Participants were clinicians (HCPs) employed in a falls prevention service or memory assessment service who were willing to take part in an interview. The researcher informed HCPs of the study at team meetings, and HCPs contacted the research team directly for further information. Snowballing was used to identify further participants, who were invited to participate through emails and face-to-face meetings. Participants were nurses (n=10), physiotherapists (n=5), occupational therapists (n=3) and a psychiatrist (n=1) working in the English East Midlands. Eleven participants worked within a falls service and eight worked within memory services. Fifteen participants were female. Participants represented a range of experience in their occupational roles. Interviews were held at a location of the participant's choice (university or the participant's place of work).

### Data collection
Interviews were conducted by a female research occupational therapist (TP) who had previous experience of working in a falls service. TP undertook training in qualitative research in preparation for conducting the interviews and was supervised by KP, an experienced qualitative researcher. Nineteen one-to-one indepth interviews took place between October 2013 and February 2014. The mean length of interviews was 46min (range 23–76). Interviews were conducted using an interview schedule, developed from researchers' knowledge of previous literature and in the context of the study objectives. They contained questions related to participants' practices, experiences of people with dementia, opinions on falls prevention and potential interventions, and the nature of the services within which they worked. There were slight differences between the schedules for participants from the falls and memory assessment services, to ensure questions were relevant to their clinical area (online supplementary appendices A and B).

### Data analysis
Interviews were audio-recorded and transcribed verbatim. Following each interview, field notes were recorded by the interviewer to gather initial impressions of the interview. Transcripts were analysed using Braun and Clarke's six-phase model of thematic analysis[28] (table 1).

Data were transcribed verbatim by a professional transcription service, checked for accuracy and independently analysed by an experienced qualitative researcher (JD). Analysis proceeded by means of a recursive process, moving between coding and reflection on the data. Regular meetings were held between the researcher who conducted the analysis, the researcher who completed the data collection (TP) and an experienced qualitative researcher (KP), who were familiar with the data. The emerging findings were discussed at each of these sessions to ensure they sat comfortably with the predominant and important issues raised by participants over the course of the study.

| Table 1 | Process of thematic analysis |
|---|---|
| **Phase** | **Description** |
| 1. Familiarisation with data | Transcripts were read and reread several times and initial patterns were noted. |
| 2. Generation of codes | Initial codes were produced manually, facilitated by handwritten notes and highlighted text. This was a recursive process, requiring rereading and recoding of data. |
| 3. Search for themes | Codes and corresponding quotes were manually collated, into handwritten lists, detailing potential themes and subthemes. |
| 4. Review of themes | The organised codes and corresponding quotes were reviewed against their candidate themes and subthemes. Themes deemed similar were collapsed into a single theme. |
| 5. Definition of themes | Themes, including subthemes, were defined and analysed in relation to the overall data. |
| 6. Production of report | Themes were embedded into a report of findings, using quotes to demonstrate each theme and subtheme. |

## RESULTS

Three main themes were identified from the data: 'challenges posed by dementia', 'adaptations to make falls prevention appropriate for people with dementia' and 'organisational barriers'. An example of how codes adhere together to form an overarching theme is outlined in table 2.

### Challenges posed by dementia

Participants recognised that dementia was associated with falls risk and that some level of risk was unavoidable.

Barriers to risk assessment and falls prevention intervention were present at an individual level. These were mostly symptoms of dementia, such as forgetfulness or poor safety awareness. The findings of this theme were clustered around three subthemes.

### Risks associated with dementia

Participants recognised that dementia is a neurological condition, causing both physical and cognitive impairments leading to falls:

> …It [dementia] isn't a disease of memory, it's a disease of the brain. There is going to be a collateral impact on people's ability to maintain their stability, their balance, to be able to react quickly to what's going on, to get their hand out to stop themselves, to realise they're putting their hand on a shadow on the wall not on the doorframe… (Occupational Therapist 2, Falls Service)

Medications used to treat dementia-related symptoms were identified as increasing falls risk. People with dementia were described as more likely to adopt risky behaviours due to impairments in insight:

> They take risks, I think…they [people with dementia] can't reason through, like the safety… (Physiotherapist 4, Falls Service)

### Acceptance of risks

HCPs felt that falls could not be prevented entirely. The effectiveness of advice and information was dependent on the patient:

> …you give them [patients] informed choices and information. And then it's up to them, and, and, you have to respect that they will make that choice themselves. (Nurse 2, Falls Service)

Despite identifying additional risk-taking behaviour as a challenge to falls prevention in dementia, participants noted that patients had the right to take risks, and where patients had mental capacity this right should be respected.

| Table 2 | Example of how codes adhere together to form a theme | |
|---|---|---|
| **Codes** | **Subthemes** | **Theme** |
| Dementia increases risk of falls | Risks associated with dementia | Challenges posed by dementia |
| Multifactorial risk factors | | |
| Medication increases risk of falls | | |
| Cannot stop falls: can reduce risks | Acceptance of risks | |
| Patient choice | | |
| Give advice to reduce risks | | |
| Lack of insight | Individual characteristics | |
| Poor memory recall | | |
| Easily distracted | | |

## Individual characteristics

In addition to increasing risk, dementia was reported to bring challenges to the falls risk assessment and therapy process. HCPs sometimes found it difficult to have an open discussion about falls with patients who had dementia. Receiving a diagnosis of dementia was considered to be a distressing time for patients, during which patients' concerns focused on the diagnosis, rather than their risk of falling:

> Falls is way, way down on their, what they're concerned about. (Nurse 1, Falls Service)

Assessment of falls risk was complicated by problems in obtaining an accurate falls history due to patients' poor recall. Participants recounted how some patients avoided discussions of falls or dementia due to fears of institutionalisation and stigma. Other patients were reported to lack insight, believing they had no memory problems or physical difficulties, and thus did not want help.

Poor recall was also a problem when employing strategies to prevent falls:

> I felt I didn't really want them to do them [exercises] on their own, because I wasn't sure whether they'd remember to do them… (Physiotherapist 2, Falls Service)

Impairments in concentration, motivation, attention, understanding and following instructions, and occasionally frustration and aggression were also identified as barriers to providing interventions.

Some of the equipment and advice usually provided to people without cognitive impairment were considered to be risky if cognitive impairment was present. For example, one participant reported:

> …there's people who accept it [walking aid] and leave it in the corner and it's just more of a trip hazard. They forget about it, so. (Physiotherapist 5, Memory Clinic)

The input and guidance of carers were believed to be essential to support the patient, without which problems could not be resolved.

> Sometimes, the problems aren't that great and could actually be sorted but because the carer can't come on board, it doesn't get sorted. (Nurse 8, Memory Clinic)

Participants sometimes revealed their own limiting beliefs around what could realistically be achieved when providing interventions for people with dementia:

> It can be frustrating. That you may not be able to make any significant changes. (Nurse 3, Falls Service)

Overall, dementia and its symptoms, such as poor recall, were reported to increase risk of falls, create a difficulty in assessing risk and cause problems with implementing interventions.

## Adaptations to make falls prevention appropriate for people with dementia

Participants identified numerous challenges to preventing falls and believed some risk had to be accepted, but they also provided a number of recommendations for how to adapt interventions when working with people with dementia. The findings in this theme were focused on participants' thoughts and views around addressing the challenges relating to risk. This large theme is clustered around seven subthemes, and included debate around the benefits and drawbacks of the various interventions for this patient group.

### Value of multidisciplinary teams

The participants recognised the multifactorial nature of falls and thus the importance of adopting a multidisciplinary approach. One participant working within the falls service emphasised:

> …the good thing about our team is having an MDT [Multi-Disciplinary Team], having physio[therapy], OT [Occupational Therapy], nursing together, is that we all approach the patient very differently. (Nurse 1, Falls Service)

### Value of seeing patients at home

Participants generally recommended that patients with dementia should be assessed and treated in their home environment. This allowed HCPs to identify potential environmental falls hazards, and to assess the patient's transfer, mobility and functional ability in the environment in which they are most likely to fall, providing the HCP with a more valid assessment:

> So, you need to look at the environment that people are in as well as other issues, you can look at the medication, and you can look at blood pressure in the clinic but you can't look at their environment that they're living in so, yeah. At home, you can incorporate all that. (Nurse 2, Falls Service)

While some participants recognised that clinic environments could be useful, transport could be a problem for patients. Family members, neighbours and friends were more likely to be present when assessing a patient at home, offering a useful source of assessment information. The home environment was also thought to encourage patients to be more honest and open, in contrast to the 'sterile clinic'.

### Groups as an intervention

HCPs recommended that interventions should take place in the home, but also recognised *"positive benefits from doing things in a group environment"* (Physiotherapist 2, Falls Service). Groups could offer social and cognitive stimulation, camaraderie, and motivation through exercising alongside others, instead of alone at home. Participants were pragmatic in recognising that groups were also more financially viable than one-to-one intervention, and thus

offered a mitigation to limited resources. Furthermore, existing local community groups could help support people to continue their exercises through:

> …introduc[ing] people to voluntary sector and community-based activities such as, the exercise and gentle keep fit group… (Nurse 10, Memory Clinic)

At the same time, groups could be problematic for people with dementia, as they may forget to attend, have difficulty travelling to groups or become distracted by others in the group. Finding groups that were able to support people with dementia of varying severity was challenging in practice. Group activities were not appropriate for everyone:

> Some like it [groups], some don't. Some are individuals. Don't want to do that. Others like speaking or talking to other people, sitting down, having a chat. (Nurse 4, Falls Service)

### Exercise as an intervention

The appropriateness of physical exercise, such as strength and balance training, as an intervention was thought to be dependent on the patient's previous level of interest in exercise:

> If they were a couch potato then [before dementia], they're probably going to be a couch potato now. (Nurse 8, Memory Clinic)

Strength and balance exercises were believed to be helpful to prevent falls. It was, however, emphasised that:

> It's about whether they can do that [performing an exercise] safely or not. (Physiotherapist 1, Falls Service)

### Recommended strategies

Suggested solutions to safety concerns included using support of family members or professional support workers when prescribing exercise programmes. This was viewed as fundamental to providing an effective falls intervention programme for people with dementia. Not all patients had the support of family members, who were earlier identified as essential sources of support for patients, and HCPs were wary of creating additional burden on family members. Participants therefore identified an important role for support workers:

> …an assistant or support worker or trained support workers can make them or encourage them to do it [prescribed exercise], that will work… (Psychiatrist, Memory Clinic)

Participants suggested various other strategies that could help people with dementia. Regular professional input, repetition, use of telecare, prompts and pictorial forms of instructions were identified as means of helping counter recall difficulties. Participants also highlighted that patients with dementia needed the intervention over

an extended period of time to enable effective learning of the new tactics or exercises. Ongoing reviews were viewed as important, as the progressive nature of dementia meant that patients may become unsafe using equipment or completing exercises.

### Catch patients early on

Participants were concerned about the patient's ability to use new equipment, and reported *"trying not to change things too much"* (Physiotherapist 2, Falls Service).

The use of mobility equipment was seen to be valuable. However, in light of concerns, it was suggested that, for patients who needed it, mobility equipment should be introduced *"earlier on in their…dementia as in, quite early on when they're still able to remember to use it [walking aid]"* (Nurse 1, Falls Service).

For any intervention method, it was believed that services should be offered at the earlier stages of dementia to achieve the best outcomes:

> So, they [people at the early stages of dementia] should be referred because that's the ideal time to go in and do the memory strategies, and talking to carers about deskilling and the importance of keeping people involved and active. (Occupational Therapist 3, Memory Clinic)

People with mild impairment were still able to learn new information and change their behaviours, while HCPs could successfully introduce new equipment, strategies and exercises to the patient and suggest changes to their environment.

### Individualised care

Participants emphasised that standard interventions should be adapted to each patient's individual interests and needs:

> I think you have your standard toolbox of things that you go through but you tailor those to that individual. (Nurse 2, Falls Service)

The success of interventions such as exercise programmes and groups depended on whether the individual was interested in them. Interventions for people with dementia were therefore recommended to be *"tailor[ed] to suit the person's individual nature"* (Psychiatrist, Memory Clinic).

Participants' expressed awareness of the diversity and effectiveness of adaptive strategies to prevent falls in people with dementia, as well as the drawbacks and challenges of implementing these in a systematic and coordinated manner.

### Organisational barriers

Despite participants recognising the challenges posed by dementia and possible interventions to address these challenges, they were constrained to operate within limited service capacity and resources, and tightly specified,

segregated, services. The findings of this large theme were clustered around five subthemes.

### Insufficient resources

People with dementia were recognised as needing more support and more time to learn new information, and to need a tailored approach. However, limited resources meant that the available support was insufficient.

HCPs reported that funds were inadequate to meet the needs of some patients:

> Our biggest difficulty actually is getting people to the exercise group safely. And that's because we don't have funding for transport. (Physiotherapist 1, Falls Service)

Increasing demand on services and inadequate availability of staff resulted in pressures to assess and discharge patients quickly:

> There obviously, (…) it is a pressure to discharge and just see new patients all the time, so, we don't get long enough, I don't think we do. (Physiotherapist 3, Falls Service)

Furthermore, although early intervention was recommended by participants, high numbers of referrals to services resulted in waiting lists, delaying intervention. This created additional problems for people with dementia:

> People have lost momentum, but they've also possibly lost a bit of cognitive skill also. So I think we need to be able to offer it [services] much swifter, and, we can't. (Nurse 7, Memory Clinic)

### Segregated services

The memory and falls services existed within tightly commissioned systems, provided by separate provider organisations, which focused on either mental or physical health needs. Services were too inflexible to meet both the complex health needs of people with dementia.

HCPs spoke of the segregation and fragmentation of these services:

> Our memory assessment service and falls service are so separate. (Nurse 8, Memory Clinic)

Participants reflected that there would be benefits from working together and sharing skills. However, this was made difficult by a lack of communication between the services, lack of knowledge of the other service, separate computer systems, a clash of understanding between the two services and having no direct working contact, preventing interdisciplinary collaboration:

> And it's taken me a long time to find out where all of falls services are and make links with the physios that work there, because we've got no way of knowing whether people have been referred to both services or more than that they can be referred to [the]

Hospital Rehab Unit, and the community falls and to me, and we wouldn't know…We're not all on the same email list even, so you've got to try and find people, so by word of mouth or phoning round. And the fact that we work for different organisations, we're not all NHS anymore, are we? (Physiotherapist 5, Memory Clinic)

### Signpost on

Pressure on memory assessment services to assess and discharge new referrals quickly, without much capacity for follow-up, led to a general tendency to 'signpost' to other services with little coordination or integration:

> Well, I suppose, in terms of the memory assessment service, we're just quite focused on memory assessment. Referrals to the CST [Cognitive Stimulation Therapy] group that I've mentioned would be a referral to a separate service that's outside our control. Referral for mobility aids and things will be to the physiotherapist which is part of the community mental health team. And referral for any other equipment or adaptations will be to the occupational therapist as part of the community mental health team. (Nurse 10, Memory Clinic)

### Identify patients with cognitive needs

The multifactorial nature of falls, impairments associated with dementia and the high rates of comorbidities among people living with dementia meant that the involvement of various HCPs and services was often required. Despite focusing on the physical needs of the patients, HCPs working in falls prevention services spoke confidently about their abilities to identify patients with cognitive impairment at an early stage:

> …even if they haven't been formally diagnosed, there is some sort of cognitive issues that you pick up initially, in the initial assessment. (Occupational Therapist 1, Falls Service)

HCPs assessed cognition as part of a falls risk assessment. Patients with cognitive impairment could then be referred on to address their mental health needs:

> We're totally separate but if the medics think that they need a memory clinic, then they will…I think, what they do is, I think they would write to the GP [General Practitioner] because I think it, think it has to come from the GP, so, the GP then, so no, we don't overlap at all. (Nurse 5, Falls Service)

### GP is central

The general practitioner (GP; family doctor) was perceived to be a 'central point' of contact, who was able to review both the physical and mental health needs of the patient, and provide a link between otherwise segregated services:

I would say, you know, go back and see your GP, you know, if it's anything physical, so that they can refer on to any appropriate agencies. (Nurse 6, Memory Clinic)

However, some HCPs reported the *"GP is also really busy and overwhelmed with work"* (Nurse 1, Falls Service), which could result in a delay in treatment. Timely treatment was important for patients whose condition was likely to deteriorate.

This theme emphasised a lack of resources and capacity within services, leading to reliance on referrals to third parties, while insufficient integration with these services could result in confusion and delay.

## DISCUSSION

This study revealed many challenges faced by HCPs when providing falls prevention interventions to people living with dementia. Dementia was recognised to increase the likelihood of falls and make falls prevention more difficult. HCPs identified barriers to falls prevention at an individual level, such as poor recall causing problems with remembering to do exercises and completing them safely, and at an organisational level. Our participants suggested ways in which interventions can be adapted to meet the needs of people living with dementia. HCPs highlighted the importance of providing falls interventions that are tailored to each patient's individual interests, which included providing opportunities to take part in interventions at home or in groups. However, HCPs were wary that home-based interventions could be costly, and thus less attractive to services, while some patients would be unable to travel to group interventions, particularly as community transport was often unavailable. HCPs recommended that falls prevention programmes for people with dementia should be delivered by a multidisciplinary team who have skills in treating people with both dementia and falls. While participants reported that they worked within a multidisciplinary team, they also described segregated mental and physical health services that struggled to communicate effectively. The GP acted as a central point of contact between the services, enabling referrals between them. However, this created disjointed and sometimes delayed care for patients.

Approximately 30% of people with dementia living in the community live alone,[29] and 16% of these have no family involved in their care,[30] so many people with dementia will not have family carer support when undertaking falls prevention activities. Some patients completing an exercise intervention have reported that they did not wish to be supervised by their spouses.[31] While carers can provide valuable support during falls interventions for people with dementia, interventions that rely on caregiver support will not be suitable for all. Professional support workers could fill this gap where they are available.

GPs are facing increasing numbers of patients and complexity of cases, with insufficient staff to cope.[32] In the UK, access to many services is typically routed through GPs. Alternative methods of accessing services are available, although this varies by service type and area. Services that are able to communicate directly, rather than relying on GPs, may be able to offer more efficient treatment.

Lower income countries and diverse cultures may experience different challenges and could offer alternative solutions to preventing falls in people with dementia. HCPs working in Malaysia previously reported that limited skills and training, and lack of informative materials to give to patients, were barriers to falls prevention in older adults.[33] This is in contrast to our study, in which HCPs presented as confident in their training and knowledge, with HCPs working within falls services reporting their capability to identify cognitive impairment, and referenced information and equipment, such as telecare, that was available for patients. Nevertheless, both studies also identified similar challenges, including difficulties discussing falls due to patient denial and feelings of stigma, ineffective communication across services, and limited staff and resources. Further research in diverse cultures, and other areas of the UK where service structures may differ, is needed to explore the various challenges and solutions to falls prevention in people with dementia specifically.

### Strengths and limitations

Qualitative studies can identify findings derived from experience and practical expertise that are difficult to ascertain in other ways. Our study focused on the perspectives of HCPs that provide services for people with cognitive impairment and older people who have fallen. The interviewer's professional experience as an occupational therapist familiar with the participants' working environment and experience may have encouraged participants to be candid and open in sharing their experiences with a fellow professional. However, this familiarity may have resulted in the failure to elicit and explore the nature of some tacit knowledge and implicit assumptions. Nevertheless, the multidisciplinary composition of the research team made alternative perspectives available for the interpretation and analysis of the data. Our participants offered an insight into organisational barriers related to limited resources and tightly defined, segregated services. Segregation of services and limited resources are common themes internationally.[16 26 33]

We report findings from a single qualitative study, involving a sample of HCPs working in falls prevention and memory services in a particular locality. The results are not necessarily transferable to other populations or services, although the results will be of interest to any community service treating people with dementia who are at risk of falling.

### Implications and future research

Participants highlighted that interventions should be adapted to individual needs and preferences, as there was no 'one size fits all' approach. This is in line with previous recommendations,[34] and is supported by other qualitative findings which suggest that patients' views and previous experiences are central to intervention uptake.[16] Recent research that

categorised different perspectives to falls and falls prevention interventions identified that some older people hold an 'ignorant perspective',[35] characterised by the belief that falls are unpredictable, of low risk and that interventions will be unhelpful. The HCPs in our study encountered patients with similar beliefs, reporting these as significant barriers to providing falls interventions. However, the HCPs interpreted these patients' attitudes to falls as a consequence of dementia, which caused poor recall of falls, a lack of insight, and resulted in patients focusing on their dementia diagnosis over future falls risk. It has been recommended that falls prevention programmes should be consistent with a positive self-identity, emphasising the benefits of participation in promoting independence and well-being, instead of the negative prospect of falls,[36] based on evidence from theory and self-reported reasons for non-participation. Individual preferences are also important to intervention adherence. In a study of prescribed exercises for people with dementia, adherence that decreased over time was explained as a result of participants favouring other exercise-based hobbies, such as walking, over prescribed strength and balance exercises.[37] Therefore, incorporating prescribed exercises into patients' individual preferred activities could increase adherence.

Segregated services that struggled to communicate effectively with each other can result in problems providing falls interventions for older people with both mental and physical health needs. HCPs generally perceived this to be a consequence of the many disciplines and different services required to address the multifaceted problems associated with dementia and falls. Some HCPs reported wanting to work more closely with other services, but did not know who to contact or how to contact them. There was a clear need for better integration of health services for older people to enable more effective communication and treatment. This could be achieved through shared electronic patient record systems, joint team meetings or a designated member of the team via which services could contact each other. Our participants spoke confidently of their ability to identify a range of health needs, including cognitive impairment. Enabling HCPs to communicate directly with one another and make referrals directly between services could reduce patient waiting times and lessen the pressures on GPs. It is important to note that this study was primarily concerned with a selection of stakeholders: HCPs working in the falls and memory assessment services. It will be important to explore the views of GPs, as well as other professionals such as practice nurses, commissioners and those working in social care and the third sector. A qualitative study has previously explored GPs' views of engaging with HCPs to prevent falls in older adults.[38] GPs similarly identified themselves as central to the referral process, and reported that communication with HCPs was important but difficult and that they were more likely to refer to individual HCPs with whom they were familiar. Further research into the views of different professionals in preventing falls specifically in people with dementia could shed more light on these issues and contribute to the formulation of recommendations for new ways of working and service restructuring.

Participants reported that additional support was required for people with dementia for falls prevention interventions to be safe and successful, but were mindful that requesting family members' input could overburden them. A study of a home-based exercise intervention that relied on informal caregiver support found that carer burden did not increase following the 6-month-long intervention.[37] However, the level of caregiver input was not controlled for, burden was not assessed throughout the intervention period, and carers volunteering to participate in an intervention with a high level of carer input may have been more motivated and less burdened than those who declined to participate.

Future research should further explore whether attitudes to falls interventions differ in people with dementia, and test whether interventions that emphasise the promotion of individuals' positive self-identities (eg, framing the intervention as a means to improve independence, well-being and functional capacity) can help the uptake of falls prevention interventions. Further research is also needed to examine the impact of exercise interventions for people with dementia on informal carers. The increasing population of older people means that limited resources will continue to be a problem,[39] although better integration and organisation of existing resources could be more cost-effective. Health and community services will likely require additional resources to address the varied needs of an increasing number of patients, and it is of particular importance for future research to investigate the cost-effectiveness of supported interventions.

## CONCLUSIONS

This study indicated that HCPs perceive that there are barriers to delivering falls interventions to people with dementia at both individual and organisational levels. They identified a number of measures which could help to overcome these. Falls interventions could be adapted for patients, for instance by conducting risk assessments and interventions at home, providing multidisciplinary input at an early stage of the disease, adapting interventions to suit the needs and preferences of individuals, and conducting intervention programmes over a longer period of time, with regular reviews and visits from support workers. These recommendations would require additional resources, but could be better enabled by the improved integration of services and empowerment of HCPs to communicate with and refer directly across different services. Future research will be needed to develop and test the effectiveness, including the cost-effectiveness, of adapted interventions in people with dementia.

**Acknowledgements** The authors would like to thank the healthcare professionals who participated in the study, and Maureen Godfrey, who was the patient and public involvement coapplicant.

**Contributors** CB drafted the manuscript. JD undertook the analysis, in discussion with TP, supervised by KP. TP undertook the interviews, supervised by KP, and

contributed to data analysis. VvdW, KP and TP designed and ran the study within which the interviews took place. LB and PL advised on study design and provided expertise on falls and memory services. RHH conceived the study, obtained funding and supervised the protocol. RHH is the principal investigator. All authors contributed to interpretation, edited the text and approved the final manuscript.

**Funding** This article presents independent research funded by the National Institute for Health Research (NIHR) under its Programme Development Grant (Reference Number RP-DG-0611-10013), https://www.nihr.ac.uk/funding-and-support/. The views expressed are those of the author(s) and not necessarily those of the NHS, the NIHR or the Department of Health and Social Care. The funders had no role in study design, data collection and analysis, decision to publish, or preparation of the manuscript.

**Competing interests** None declared.

**Patient consent for publication** Not required.

**Ethics approval** Ethics approval was obtained from the NHS National Research Ethics Service Committee, East Midlands (Reference 13/EM/1061). Information sheets were sent to potential participants and written consent was obtained before the interview. Participants were informed they could withdraw from the study at any time; however, no one did. All participants gave informed consent to participate.

**Provenance and peer review** Not commissioned; externally peer reviewed.

**Data sharing statement** Data are not routinely available, but interested researchers should enquire with RHH.

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
