## [Reviewer comments · BMJ Open]

This paper was submitted to a another journal from BMJ but declined for publication following peer review. The authors addressed the reviewers' comments and submitted the revised paper to BMJ Open. The paper was subsequently accepted for publication at BMJ Open.

(This paper received three reviews from its previous journal but only two reviewers agreed to published their review.)

ARTICLE DETAILS

TITLE (PROVISIONAL)	Perspectives of healthcare professionals in England on falls interventions for people with dementia: A qualitative interview study
AUTHORS	Burgon, Clare; Darby, Janet; Pollock, Kristian; van der Wardt, Veronika; Peach, Tamsin; Beck, Lyndsay; Logan, Pip; Harwood, Rowan

VERSION 1 – REVIEW

REVIEWER	Dr. Nadine Pohontsch University Medical Center Hamburg-Eppendorf, Germany
REVIEW RETURNED	06-Aug-2018

GENERAL COMMENTS	thank you very much for your manuscript on this important topic. I really enjoyed reading it and will recommend it to be accepted after minor revisions suggested below. The introduction is clearly structured and the research question adequately deducted and phrased, but I miss a statement on whether research already exists that tries to answer your or related research questions. The methods are described in a clear and comprehensible way. Could you just add some information on the interviewer's experience in qualitative interviewing techniques or training received? Please also add information on software used, if any. Could you please also elaborate shortly on your interviewees' gender, age and years of work experience and whether interviewees received their transcripts for checking/commenting or the results for providing feedback. The results section is easy to read and findings are adequately supported by participants' quotes. The Strength and Limitations section could profit from discussing the possible bias introduced in the data by participants self selection. Are the participants recruited from one or more than one fall service/memory service? If they are working just in one service respectively, how could this have possibly biased the data? Overall, the discussion and conclusion is appropriate. I have some more minor comments: Could you include the completed COREQ checklist (as suppl. material)? line 73: Representativity is not a quality criteria for qualitative research. Transferability/generalisability are more appropriate concepts. Please change the wording here to avoid confusion with
--

	statistical representability which is not achievable in a qualitative study. line 155: Could you present the references of the literature used here? How were expert opinions collected? line 168: Could you give more information on the peer debriefing process? line 464-467: This statement is not unique in respect to your study. Could you elaborate more on what you did to reduce this problem and whether you think that you were successful? Again, thank you very much for that interesting manuscript.
--	--

REVIEWER	Wah Yun LOW Faculty of Medicine, University of Malaya, Kuala Lumpur, Malaysia
REVIEW RETURNED	09-Sep-2018

GENERAL COMMENTS	This study is very time and appropriate in view of the escalating injury issues pertaining to the elderly and the increase in population of the aging society. The study aimed to explore the experiences of healthcare professionals working in falls prevention and memory assessment services of providing assessments and interventions for falls risk reduction in people with dementia. My comments are as follows: (1) In the Abstract, please include how many IDI (in-depth interviews or focus group discussions) that were carried out on these various participants (2) In page 6, in total, how many in-depth interviews (IDI) and focus group discussion (FGDs) were carried out of these participants. Were these just one-to-one interviews? If so, in order to triangulate the data, why wasn't focus group discussions carried out? Was saturation achieved with these number of participants? What are the inclusion and exclusion criteria in selecting these participants? Would be good to have a table showing the socio-demographic characteristics of the healthcare provider as well as the dementia patients (e.g. age, gender, years of working, duration of dementia, staying alone, etc, etc) of these participants to be included in the Results section (2) In page 8, what method of coding was used? Perhaps you can show one example of how one particular theme was derived based on the coding method. show how these themes emerged from the codes (3) In page 9, was the next-of-kin or caregiver interviewed?-- to triangulate the data on the recall of falls history (4) In page 16, the results clearly show little coordination on communications between departments/division providing healthcare services. With this, what measures are recommended to trouble shoot these problems (5) In the Discussion, perhaps the existing practice(c) on helping dementia patients with fall prevention needs to be discussed (6) Please elaborate how "diverse cultures" faces different challenges in fall prevention among patients with dementia
---

REVIEWER	Alison Wheatley Institute of Health and Society, Newcastle University, UK
REVIEW RETURNED	10-Sep-2018

GENERAL COMMENTS	This is an interesting paper which addresses the important issues of fall prevention services for people with dementia. The study is well-designed, with methodology appropriate for the research question posed.
--

	I do, however, have a number of queries and comments:  1) Pg 5, line 106 describes stakeholders in this context as 'people living with dementia, their families and other carers, and healthcare professionals'. To what extent were social services and the third sector (who often commission relevant services such as telecare and reablement as well as community falls prevention programs) considered in this study? 2) Pg 6, line 149 - the characteristics of the interviewer are described, but there is no reflection on the potential impacts on the data. 3) Pg 7, line 155 – could you elaborate on what 'expert opinion' means in this context? 4) I found the framing of the results to be slightly confusing. The first two themes, 'Challenges posed by dementia' and 'Adaptations to make falls prevention appropriate for people with dementia' suggest a link from one to the other (i.e. adaptations made based on challenges identified). However, these links were not always clearly stated. I wonder if this is because professionals were unable to identify appropriate adaptations for every challenge posed? If so, I think this should be reflected upon in the discussion. Otherwise, some clearer signposting could help lead the reader through the results.
--	---

REVIEWER	Areeba Kara Indiana University Health Physicians and Indiana University School of Medicine. Indianapolis, Indiana, US
REVIEW RETURNED	23-Oct-2018

GENERAL COMMENTS	The authors have undertaken a qualitative study to describe the perspectives of clinicians caring for patients with dementia on fall prevention. As patients with cognitive impairment are often excluded from fall prevention investigations, this topic is important. There are several issues that need clarification:  1- Please explicate the setting of your work. Specifically, for readers outside the UK (like myself), it is unclear how the therapists and GPs interact. A few lines describing the structure would therefore be helpful. Similarly, please clarify in the introduction that your focus is on outpatient fall prevention. 2-Details on recruitment of participants and then basic demographics of participants are missing: were they all from the same center? Age/Gender/Experience (if available)How many were eligible and declined to participate.?It is notable that there are no GPs represented in the sample. Please explain if they had an opportunity to participate which they did not avail vs a different scenario. This is a major limitation as if GPs anchor the process, their view would be important to include in any work gathering information to redesign systems. 3- Please include and complete all relevant items in the COREQ checklist 4- The three main themes are rich in information. Please consider sectioning each theme into sub themes with sub headings to make the information easier for readers to grasp. For e.g the adaptation theme may lend itself to a theme of ' in home vs group' , social support etc. The first and second theme both appear to have a sub theme of patient autonomy embedded in them. My concern is that valuable information in the themes will be missed without some reorganization. 5- I am confused by the perspectives offered in lines 209-213. These appear to be patient views and not care provider views.
--

	6- As one of the stated objectives of the work was to elicit recommendations for designing best practice strategies for fall prevention in patients with dementia there seems to be a missed opportunity in the discussion to lay out some sort of blueprint for this based on your results. While there are certainly recommendations embedded in the discussion, perhaps more general guiding principles will be helpful. E.g. the system would need to be flexible to the stage of dementia, availability of care givers, interprofessional etc. Thanks for the opportunity to review your work! All the very best!
--	--

VERSION 1 – AUTHOR RESPONSE

Reviewer 1.

To our knowledge this is the first study that has explored the perspectives and experiences of health care professionals of falls intervention for people with dementia. We have provided further information regarding the interviewer’s training in lines 146-148. In small scale qualitative studies of fewer than 30 participants it is often recommended that analysis is conducted by hand as it provides a greater level of intimacy and familiarity with the data. Therefore data were analysed manually (i.e. not using software). Further clarification has been added to table 1 which appears in line 162. See lines 136-137 for details of gender and experience of participants. Participants often reported their years of experience in their role, though this was not systematically asked. Age of participants was not collected. We did not undertake member checking. We discuss the limitations of recruiting participants from one locality in lines 495 -497.

Minor comments:

“Line 73:” We have changed the wording (see line 71)

“Line 155:” Expert opinion refers to the experienced researchers' knowledge of previous literature. This has been clarified in line 151

“Line 168:” We have outlined the process of peer debriefing in lines 167-171.

“Line 464-467:” We have decided to remove this statement from the paper as this is the inherent nature of qualitative research in which interviews produce situated accounts which may include shifting and conflicting perspectives.

Reviewer 2.

1) We have now included the number of interviews in line 41.

2) The study was designed as one-to-one interviews with health care professionals, which we have clarified in line 148. The study was not based on grounded theory methodology. Consequently, the concept of saturation was not relevant. A sample size of 20 professional respondents was determined at the start of the study. This was chosen as it has been judged in qualitative research to constitute a sufficiently wide range and diversity of perspectives to enable understanding of the topic under investigation. Inclusion criteria have been clarified in lines 119-122. Additional demographic information about the participants has been included in lines 136-137. Participants often reported their years of experience in their role, though this was not systematically asked. Age of participants was not collected.

(‘2’) We have provided an example of how codes formed themes in table 2, line 177

3) This study reports results of interviews with healthcare professionals only. We have previously reported on the results of interviews with people with dementia and their next-of-kin / caregivers, which we refer to in lines 106- 107.

4) We have made recommendations to troubleshoot the problems of communication in lines 526-531

5) Existing practice on fall prevention has been set out in the introduction

6) We have provided an example where a different culture experienced different challenges in older adults generally (lines 470-478). We have now elaborated that further research is needed to explore this with people with dementia specifically (lines 478-480)

Reviewer 3.

- 1) A consideration on the lack of involvement of other stakeholders is now included in the discussion (lines 531-541)
- 2) We have discussed the potential impact of the interviewer's professional background on the data in lines 484-491
- 3) Expert opinion refers to the experienced researchers' knowledge of previous literature. This has been clarified in lines 150-151.
- 4) We have introduced subheadings to emphasize the subthemes within the three main themes, and included additional text throughout to link these themes together

Reviewer 4.

- 1) We have elaborated on the service structure in lines 465-466 and the out-patient focus in lines 101 & 113
- 2) Additional demographic information about the participants has been included in lines 136-137. Participants often reported their years of experience in their role, though this was not systematically asked. Age of participants was not collected. The coordinating role of GPs became clear from this study's results. We have added this to the discussion (lines 531-541). Participants often reported their years of experience in their role, though this was not systematically asked. Age of participants was not collected.
- 3) We have included the SRQR checklist instead, as per the editor's request
- 4) We have introduced subheadings to emphasize the subthemes within the three main themes, and included additional text throughout to link these themes together.
- 5) We have now clarified that this was the HCPs report on their experiences of patients in lines 217-219.
- 6) We have made recommendations in lines 517-518; 526-531; 557-560; 564-571. Further work will be required to formulate a blue print of recommendations (see lines 538-541).

VERSION 2 – REVIEW

REVIEWER	Nadine Pohontsch Department of General Practice / Primary Care University Medical Center Hamburg-Eppendorf
REVIEW RETURNED	06-Dec-2018

GENERAL COMMENTS	Thank you for your revision. No further comments.
---

REVIEWER	Alison Wheatley Newcastle University, UK
REVIEW RETURNED	17-Dec-2018

GENERAL COMMENTS	In my opinion, the authors have adequately addressed my concerns and those of my fellow reviewers. The structure of the results section in particular is significantly improved.
--